# From Bioinspired Glue to Medicine: Polydopamine as a Biomedical Material

**DOI:** 10.3390/ma13071730

**Published:** 2020-04-07

**Authors:** Daniel Hauser, Dedy Septiadi, Joel Turner, Alke Petri-Fink, Barbara Rothen-Rutishauser

**Affiliations:** 1Division of Surgery & Interventional Science, Royal Free Hospital, University College London, London NW3 2PS, UK; joel.turner.17@ucl.ac.uk; 2Adolphe Merkle Institute, University of Fribourg, 1700 Fribourg, Switzerland; dedy.septiadi@unifr.ch (D.S.); alke.fink@unifr.ch (A.P.-F.)

**Keywords:** polydopamine, biomedicine, mussel-inspired, nanomaterials

## Abstract

Biological structures have emerged through millennia of evolution, and nature has fine-tuned the material properties in order to optimise the structure–function relationship. Following this paradigm, polydopamine (PDA), which was found to be crucial for the adhesion of mussels to wet surfaces, was hence initially introduced as a coating substance to increase the chemical reactivity and surface adhesion properties. Structurally, polydopamine is very similar to melanin, which is a pigment of human skin responsible for the protection of underlying skin layers by efficiently absorbing light with potentially harmful wavelengths. Recent findings have shown the subsequent release of the energy (in the form of heat) upon light excitation, presenting it as an ideal candidate for photothermal applications. Thus, polydopamine can both be used to (i) coat nanoparticle surfaces and to (ii) form capsules and ultra-small (nano)particles/nanocomposites while retaining bulk characteristics (i.e., biocompatibility, stability under UV irradiation, heat conversion, and activity during photoacoustic imaging). Due to the aforementioned properties, polydopamine-based materials have since been tested in adhesive and in energy-related as well as in a range of medical applications such as for tumour ablation, imaging, and drug delivery. In this review, we focus upon how different forms of the material can be synthesised and the use of polydopamine in biological and biomedical applications.

## 1. Introduction

In recent years, nanomaterials have gained considerable interest [1,2]. Due to their intriguing characteristics, several applications in different fields could be advanced [2]. Such nanocharacteristics include size, possibility for surface modification, ability to carry an active payload, and photochemical/physical activity (light absorbance and fluorescence generation) [3]. The field of nanomedicine in particular has grown steadily, despite the low translation rate from bench to bedside [3,4]. Novel ideas to mimic (nano)structures inspired by nature could provide new solutions and accelerate research in this area [5,6,7]. A prime example of this is polydopamine (PDA). PDA was first identified to play a crucial role in the adhesion of sessile mussels and has gained increased attention throughout the last decade [8]. After initial use in coatings for bulk materials, it was also investigated for applications in nanotechnology. Not only was PDA used to coat nanomaterials, but it also had the ability to be synthesised into nanoparticles (NPs) [9,10]. This review will highlight the different applications for which PDA has been investigated. In the first chapter, bioinspiration discovery from mussels and humans leading to the development of polydopamine as functional platform is highlighted (Figure 1A). Then, theoretical and experimental proof of the chemical formation of polydopamine-based structures is summarised (Figure 1B,C). This is then followed by descriptions of the unique (physicochemical) properties of polydopamine. Their applications as adhesive and in energy-related applications are also highlighted (Figure 1D). Throughout, their wide range of applications in the biomedical field (thanks to their biocompatibility) is thoroughly explained. This includes examples of theranostic and drug delivery applications derived from different types of polydopamine-based configurations: adhesive, particle coating agent, capsule, and NPs/composites, respectively (Figure 1C,E). Lastly, some challenges to make PDA into functional tools for future clinical applications are envisaged.

## 2. Bioinspiration: From Mussel to Human Skin

### 2.1. Bioinspiration by Mussels

The observation of the ability of mussels to adhere to various wet surfaces strong enough to withstand strong ocean currents has sparked investigations into the role of PDA in this property (Figure 2A) [11,12]. Mussels were found to adhere to surfaces using so-called ‘byssi’ (Figure 2B). Byssi consist of multiple threads with a length of 2–6 cm, each featuring an adhesive plaque, a flexible proximal portion, and a more rigid distal portion [13]. The threads are formed during two distinct stages. In the first stage, the animal identifies a suitable substrate for adhesion by placing a protrusion from inside the living space, called a mussel ‘foot’ onto the substrate, which triggers the internal assembly of essential molecular components [13]. The second stage is the disengagement of the foot while the thread is attached firmly to the surface [13]. This process can last several minutes in adult mussels and can be as fast as 30 s in juvenile mussels [14]. The threads are produced individually inside a confined compartment of the mussel foot, which is referred to as the ‘ventral groove’ [15]. Subsequently, the so-called phenol, collagen, and accessory glands then secrete defined amounts of the contents into the ventral groove [16]. The adhesion itself is mediated by the dopamine-containing adhesive plaque (although this exact molecular process is still poorly understood) (Figure 2C–E). However, previous studies have led to a closer investigation of the physicochemical properties of the material resulting in a plethora of different approaches to translate these fantastic properties into modern biomedical applications [17,18].

### 2.2. Bioinspiration by Human Skin

Another interesting physicochemical property of PDA is its ability to absorb visible light and convert it into heat [19]. This heat generation is highly efficient, with about 99% of photon energy conversion [20]. While the exact process remains elusive, it is hypothesised to be based on the structural relation of PDA to the naturally occurring polymer melanin [21]. There are two types of melanins produced in the human skin: reddish-yellow pheomelanins and brown-black eumelanins [22]. They differ in structure and composition, with eumelanin being more closely related to PDA [23]. In general, melanin is produced by so-called melanocytes residing between the dermis and epidermis [24]. Melanocytes are highly interconnected with keratinocytes in the epidermis, to which they transport produced melanin via extracellular dendrites [24]. Interestingly, the number of melanocytes is virtually independent from skin colour, with differences in skin tones due to the random distribution of melanin. In fair skin, the melanin-containing vesicles (i.e., melanosomes) are clustered above the nuclei for the maximal protection of DNA, whilst in darker skin, the melanosomes are more evenly spread [25]. Upon UV irradiation, the skin gains increased photo protection properties. This reaction is commonly known as tanning [26,27]. There are two temporally distinct reactions to UV light exposure: one immediate and one that takes several weeks [28,29]. The immediate reaction comprises the polymerisation and redistribution of already present melanin (occurring in seconds to minutes) [29]. On the other hand, delayed tanning reactions occur in a timeframe of several hours. This process involves a complex biochemical progression that ultimately results in the activation of melanocytes and the production of melanosomes [29]. The darkness of skin has been correlated with a significant (15–30 fold) decrease in the risk of skin cancers [30,31]. In darker skin tones, the lower epidermis is better protected against DNA damage by UV light through a combination of melanin production, redistribution, and more efficient removal (apoptosis) of UV-damaged cells than in fair skin [26,32]. In this case, harmful UV irradiation is absorbed by melanin, and the energy is released again in the form of heat [33]. Dissipating heat is advantageous over UV light because of its significant lower tissue toxicity, which is thought to be responsible for the evolutionary development of melanin pigments [34]. This is similar to eumelanin, its structural relative PDA, which also exhibits low toxicity towards tissue [23,35].

## 3. Formation of PDA

Even though the exact mechanism of PDA formation is still under investigation, some aspects of the process are known. For instance, under oxidative conditions, dopamine polymerises to PDA [36]. This is a multi-step process, of which only the initial steps have been proven experimentally [37]. It is assumed that the synthesis mechanism is similar to the biosynthesis of melanin [38,39]. The first step of the reaction under basic conditions is the oxidation of dopamine to dopamine-quinone [37]. Then, the dopamine-quinone reacts via intramolecular cyclisation (1,4 Michael-type addition) to leucodopaminechrome, which oxidises further and is rearranged to 5,6-indolequinone [37]. According to the melanin-inspired model of synthesis, the following steps involve dimer and eventually oligomer formation by 5,6-dihydroxyindole and 5,6-indolequinone [40]. In contrast to the model of a covalently branched polymer (i.e., melanin-inspired model), Dreyer et al. reported a reaction mechanism resulting in a polymer consisting of non-covalent interactions such as charge transfer, hydrogen bonding, and π–π stacking [41]. While nitrogen-containing cyclic structures (i.e., indolines and indoles) support the melanin-inspired model, saturated indolines, which were produced instead of unsaturated ones, further favour the alternative mechanism by Dreyer et al. [41]. Finally, a third possibility was proposed: Both of the mechanisms above take place at the same time (Figure 3) [42]. This hypothesis was based on the fact that 5,6-dihydroxyindole, together with unpolymerised dopamine, can form a stable, self-assembled complex [42]. Moreover, one unit of dopamine together with two units of 5,6-dihydroxyindole form stable trimers, which were found incorporated into the final polymer [42]. The polymerisation of dopamine and coupling between PDA and block copolymers can be performed, (denoted as the polymerisation-coupling process) and it can induce the self-assembly of block copolymers to yield ordered structures, including micelles and vesicles [43].

## 4. Physicochemical Properties of PDA

As already mentioned, melanin and PDA exhibit a very similar structure and the physicochemical properties are largely similar [44]. One is the absorption of light: PDA’s absorption lies over the whole visible spectrum but has its maximum in the UV region [21]. Light irradiated upon PDA is very efficiently (and non-radiatively) converted into heat [45]. This particular profile of absorption can be attributed to the structure of PDA [45]. The synthesis includes the oxidative reaction of dopamine into dopaindole and dopachrome, of which both show pronounced absorption in the UV part of the spectrum [46]. However, not only the absorption of light and heat-conversion but the adhesive properties displayed by PDA have sparked great interest in the material. It is now commonly accepted that the outstanding adhesion properties can be attributed to the catechol groups present in the polymer [47,48,49]. PDA can undergo various reactions with different surfaces. Thiol and /or amine functional groups containing surfaces can—under oxidative conditions—react covalently with dopamine through Michael addition and/or Schiff base reactions [8]. Without the presence of these functional groups PDA can interact with the surfaces via hydrogen bonding, chelation, metal coordination, or π–π stacking [8]. In addition to the adhesive properties, PDA displays a variety of functional groups and can react in different ways: While species containing amine groups would react in a Schiff base reaction, a Michael addition reaction would take place with thiol-containing molecules [8]. The outstanding adhesive properties in combination with the straightforward chemical modification possibilities of PDA catalysed the investigation of PDA as a coating for inert surfaces to increase their chemical reactivity.

## 5. PDA for Adhesive and Energy-Related Applications

One of the very first applications of PDA was the application as a functional surface. Lee et al. produced PDA coated polymer pillars that could be used as a biological adhesive. In fact, this coating was reported to improve the adhesive (substrate to substrate) properties 15-fold [50]. In addition to the increase of strength, the durability of the interaction increased significantly. Even after 1000 cycles of contact (in both dry and aqueous environments), the adhesive properties did not deteriorate [50]. Lee et al. further reported the PDA deposition on almost any surface by simply immersing it into a buffered solution containing dopamine. Then, this treatment was demonstrated to increase grafting possibilities [17]. For instance, enzymes, other polymers, polysaccharides, proteins, DNA, and NPs have been grafted onto a PDA coating up to date [17,51,52,53,54,55]. For example, Ma et al. used the adhesive properties of PDA to increase the binding efficiency of environmental pollutants on clusters of superparamagnetic iron oxide NPs [56]. Another example is the use of PDA as an additive in hydrogel composites. Due to the abundant catechol groups cross-linking with Fe^3+^ ions occurs, this is similar to the function in the mussel byssi. This was further demonstrated by Lu and co-workers, who modified clay with a layer of PDA [57]. Then, the modified clay particles were introduced into epoxy resin, leading to improved dispersion of the particles and increased interfacial stress transfer, overall improving the features of the material [57]. Nam et al. used PDA to improve dye-sensitised solar cells by coating titanium oxide (TiO_2_) layers with PDA [58]. The coated dye-sensitised solar cells were found to perform with a power conversion efficiency of 1.2% [58]. PDA was also investigated for their use as a solar fluid dopant in the generation of solar thermal energy as an alternative to photovoltaics [59]. To this end, PDA nanoparticles, consisting of PDA and bovine serum albumin, were added to a so-called solar fluid whose efficiency was then assessed in a 3D-printed flow circuit using a solar simulator as an irradiation source [59]. It was found that the addition of the NPs indeed increased the efficiency of the solar fluid up to 280% compared to the plain solar fluid [59]. PDA NPs containing bovine serum albumin (BSA) (bPDA/BSA) exhibited superior heating abilities of the solar fluid in comparison to micron-sized soot particles (food colorant) or un-supplemented solar fluid (Figure 4A). Nano-sized PDA/BSA (sPDA/BSA) NPs on other hand also displayed an increased heating ability of the solar fluid in comparison to silver NPs or plain solar fluid consisting of a mixture between glycol and water (Figure 4B). The authors also confirmed that the calculated specific heating rate of PDA NPs was size-dependent (Figure 4C). In a recent publication by Zhang et al., PDA was also investigated as stabilisers for dextran and poly(ethylene glycol) (PEG)-based aqueous emulsions [60]. The synthesised capsules remained intact after surfactant addition, and upon dilution with water, only minor swelling of the droplets was observed, demonstrating the enhanced stability of the emulsion. The extraordinary properties of PDA have helped to improve various technologies. However, because of its properties such as the biocompatibility, biodegradability, and versatility, PDA is even more heavily investigated for applications in biomedicine.

## 6. Biomedical Applications of PDA

### 6.1. PDA as a Medical Adhesive

Medical adhesives have gained increasing attention as a tissue sealant, although some considerable challenges, such as weakened adhesion in physiological milieus or low biocompatibility, are still reported [61,62]. Hence, inspiration to overcome these shortcomings came from the animal kingdom, and investigations on how marine animals adhere onto mineral or biological surfaces have been performed. The progress of this field of research has recently been reviewed by the Messersmith group [63]. Mussels are one example of marine animals that inspired humans to synthesise various materials as a medical adhesive exploiting the mechanism of how their byssi adhere to substrates.

#### 6.1.1. Biocompatibility of PDA as a Medical Adhesive

Due to the applications of PDA in the context of medical adhesives being manifold, and sometimes even only parts of PDA’s adhesive mechanism (i.e., catechol coordination of its precursor) are used, every new material’s biocompatibility must be investigated separately. The biocompatibility of PDA as a material will be explained in detail in following chapters. 

#### 6.1.2. Biomedical Applications of PDA as a Medical Adhesive

In order to functionally adhere to tissue, the adhesive should be able to interact with the surface of the tissue. It was found that cysteine and lysine residues are very commonly accessible; hence, their interaction with the tissue of interest should result in optimal adhesion [64]. Although the exact mechanism remains elusive, it is known that o-quinone (an oxidation product of catechol, which is abundantly present in PDA) readily reacts with the amino acids present on the tissue [13]. Benedict et al. produced a polypeptide containing l-3,4-dihydroxyphenylalanine (L-DOPA, a precursor of dopamine), which exhibited an adhesive strength of 32 kPa on bovine corneal tissue [65]. Messersmith and co-workers used a DOPA-motif functionalised poly(ethylene glycol) (PEG) macromere and reported an adhesion strength of 35.1 kPa on porcine dermal tissue. This corresponds to a fivefold increase in comparison to fibrin-based adhesives [66]. Then, the material was also used in mice, where no signs of adverse effects were shown, and the transplanted islets were held in place for a prolonged time (i.e., one year) while reversing the diabetic phenotype [67]. Haeshin and Cho Lee oxidised dopamine-conjugated hyaluronic acid with sodium iodate (NaIO_4_) in a hydrogel and reported an excellent strength of adhesion of 48 kPa onto heart tissue [68]. Fan et al. published an L-DOPA functionalised and cross-linked (using genipin or Fe^3+^) hydrogel that showed adhesive forces of 194 and 25 kPa on porcine cartilage tissue and porcine dermal tissue, respectively [69]. The examples listed here used dopamine, or its biological precursor L-DOPA, to improve the properties of the materials. In the following chapters, the potential of PDA as a material, as opposed to its building blocks, will be discussed.

### 6.2. PDA Coatings of (Nano)Materials

PDA has been used to coat (nano)materials to improve and/or change their properties. The coating process is straightforward and universally applicable. To coat a (nano)material with PDA, it is sufficient to immerse materials of interest in a dopamine solution [17]. Messersmith et al. first reported this immersion process. For example, materials that are submersed in a solution of dopamine with a concentration of 2 mg/mL in 10 mM tris(hydroxymethyl)aminomethane (TRIS) at pH 8.5 showed a deposited PDA film with a thickness of 50 nm [17]. These reaction conditions are still the primary go-to protocol for the formation of PDA films [70]. Moreover, they can be employed for bulk materials as well as for nanomaterials. There is also the possibility to spin-coat PDA onto surfaces; however, this approach is predominantly used for electronic applications, and thus the relevance for the biomedical field is marginal [71,72].

#### 6.2.1. Biocompatibility of PDA-Coated (Nano)Materials

PDA is an inert material and has been shown to be non-toxic to living matter once being used as a coating material [36,73]. Various in vitro studies using different cell types (i.e., osteoblasts, fibroblasts, neurons, and endothelial cells) were performed by Ku et al. An improved adhesion upon cell contact with polytetrafluoroethylene surfaces functionalised with PDA compared to surfaces without PDA was observed. Additionally, there were no detrimental effects on the cells found [74].

#### 6.2.2. Biomedical Applications of PDA Coatings

The ability of bacteria and fungi to adhere to surface coatings is a major problem in food industry or medicine. The challenge lies in making surfaces that can prevent or enhance the pathogen adhesion, since a stronger bacterial adhesion can also be used for different applications [75]. The advantages of a PDA coating for such an application were investigated by Liu et al. [76]. A soft bilayer actuator was coated with PDA, increasing the adhesion of bacteria 10-fold as compared to the uncoated actuator, greatly facilitating the accumulation of bacteria in physiological media [76]. Elimelech and co-workers used the same principle to immobilise a single bacterium on a PDA-coated atomic force microscopy cantilever [77]. This approach has the advantage that the immobilised bacterium was still metabolically active [77]. Fungal infections are one of the highest incidence rate complications in hospitals [78,79]. Paulo et al. used silica NPs conjugated with amphotericin B, which is an approved antifungal compound, and immobilised the particles onto glass by an adhesive PDA layer [80]. The modified glass was found to be neither haemolytic to red blood cells nor have a cytotoxic effect on mononuclear cells but indeed exhibited contact-mediated antifungal activity. The NPs were found to be useful in supporting the continuous battle against fungal infections as a major healthcare concern and could be used as a coating or in suspension [80].

PDA coatings also have been used for biosensing applications [75]. For instance, Liu et al. produced nicotine-imprinted sensors made of PDA [81]. The PDA biosensors were able to bind 98% of 5 μM nicotine dissolved in human serum. Wang and co-workers produced haemoglobin-imprinted, PDA-coated superparamagnetic iron oxide NPs (SPIONs) [10]. In a competitive binding assay with 5 different (i.e., non-imprinted) proteins, the binding efficiency for haemoglobin was always over 80% [10]. This suggests a high potential for the use of PDA-coated NPs in biosensing. However, a major challenge in this field is the immobilisation of functional enzymes on surfaces [75]. PDA—together with gold NPs—were used to entrap glucose oxidase on a carbon electrode [82]. This electrode was reported to have superior features in terms of detection limits, long-term stability, linearity, and sensitivity when compared to a similar approach using chitosan [82]. The biosensor was successfully used to detect glucose in diluted human serum, indicating its potential for clinical use [82]. PDA coatings were also used by Lin et al. to coat SPION clusters for a combination of the detection of mRNA and photothermal therapy [83].

Apart from diagnostics, PDA coatings are also used for various approaches to treat cancer. In general, regarding the non-specific nature of cancer treatments, both healthy and diseased cells are often damaged, causing severe side effects in patients. Thus, drug delivery vectors have become one of the most promising applications of NPs in the biomedical field with the aim to directly target cancer cells and attenuate side effects [84,85]. Due to PDA’s optimal characteristics (i.e., biocompatibility, versatility, and biodegradation), the material is currently used extensively for drug targeting. For instance, Park et al. coated poly(lactic-co-glycolic acid) NPs with PDA to enable the grafting of ligands onto the NPs to enhance cellular uptake [86]. Another drug delivery approach was published by Zong et al. [87]. The researchers coated 5-fluorouracil-carrying liposomes with PDA (Figure 5) [87]. The PDA coating kept the molecule inside the liposomes until the liposomes became permeable in an acidic environment, resulting in the release of the payload [87]. Chang et al. chose a similar approach, but instead of liposomes, the researchers modified mesoporous silica NPs with PDA in order to control the release of a loaded drug by low pH [88]. The NPs were found to be internalised by cells and were reported to successfully deliver their cargo, with PDA as a gatekeeper [88].

Additionally, by coating iron oxide (Fe_3_O_4_) surfaces with PDA, or the absorption of metal ions (i.e., Fe^3+^, Fe^2+^, Gd^3+^ and Mn^2+^) into PDA NPs, PDA can be used for magnetic resonance imaging. Li et al. employed electron paramagnetic resonance, magnetometry, and nuclear magnetic relaxation dispersion to characterise PDA NPs loaded with iron [89]. The results indicated the presence of isolated iron centres with paramagnetic properties within the NPs, resulting in the measured magnetic resonance imaging signals [89]. Similarly, magnetic NPs were coated with PDA as described by Mrowczynski et al. [90]. The imaging properties were complemented by loading the composites with doxorubicin, and it was shown that the particles delivered the drug efficiently in vitro [90]. Priestly and co-workers also synthesised PDA-coated iron oxide NPs and loaded the anticancer drug bortezomib into the PDA shell and additionally doped them with gold NPs [91]. Then, the researchers used the nanoparticle system as a catalyst and, after having converted the PDA layer into carbon), also successfully used it as an adsorbent to remove rhodamine B from a solution [91].

In addition, gold NPs can be modified with a PDA. Yao and co-workers produced PDA-coated gold NPs functionalised with a bovine serum albumin–dextran conjugate [92]. Then, the NPs were used for computer tomography imaging and tumour ablation [92]. Due to the dextran brush surface, the blood circulation time could be prolonged to a point where intravenous injection with subsequent photothermal treatment of a tumour became feasible, as enough of the nanoparticle had accumulated at the tumour site [92]. Zeng et al. combined targeting with magnetic resonance imaging/computed X-ray tomography, dual-mode imaging, and photothermal therapy using gold NPs coated with PDA [93]. However, the photothermal effect in this case was not mediated by PDA but rather by the addition of indocyanine green [93]. Then, the nanoparticle surface was modified further by the self-assembly of gadolinium-1,4,7,10-tetraacetic acid and lactobionic acid onto their surface [93]. The surface-modified nanoparticle was reported to be selectively internalised by liver cancer cells, and the photothermal feature was also confirmed in vitro [93]. Lin et al. produced PDA-coated Cu(II)-doped gold nanorods, thereby prolonging their blood circulation time and increasing their photothermal performance while simultaneously reducing the toxicity of pristine gold nanorods [94]. The modified and biocompatible nanorods were found to enable computer tomography imaging, magnetic resonance imaging and exhibit chemotherapeutic functions by inhibiting tumour growth [94]. In addition, more complex gold NPs have been coated with PDA, i.e., gold nanostars [95]. These particles show optimal optical properties and are an excellent system for theranostic applications [95]. By the deposition of a PDA layer onto the surface of the gold nanostars, the light to conversion efficiency could be significantly improved, which was shown in vivo and in vitro by computer tomography imaging and photothermal therapy of cancer [95].

Light-responsive nanoprobes for safe and efficient phototheranostics consisting of PDA-coated gold nanobipyramids conjugated with doxorubicin drug were successfully synthesised by Liu et al. [96]. The probes not only exhibited higher photothermal conversion efficiency (42.07%) and stronger photoacoustic signal than those of gold nanoparticle controls, but also possessed dual-responsive doxorubicin release upon pH and photothermal stimulation. An in vitro experiment using 4T1 cells showed that cell viability was reduced to about 5% when a low concentration of nanoprobes (60 μg/mL) and low-dose laser irradiation (1.0 W/cm^2^) was applied. By modelling 4T1 tumour-bearing nude mice, an in vivo photoacoustic imaging experiment was successful, and the tumours were completely inhibited, representing the synergistic effect of photothermal treatment and chemotherapy.

In addition to gold and iron oxide NPs, other types of nanoparticle have been modified with PDA as well. For instance, Mei and co-workers designed NPs consisting of PDA surface-modified D-α-tocopherol polyethylene glycol 1000 succinate-poly(lactide) particles loaded with docetaxel and functionalised with galactosamine to target cancer cells in the liver [97]. The NPs were reported to significantly inhibit cancer growth in vitro and in vivo even more successfully than a clinically approved formulation of docetaxel [97]. Yu and co-workers coated Prussian blue NPs with human serum albumin and PDA and loaded them with doxorubicin [98]. The release of doxorubicin from the NPs can be triggered by pH or near-infrared irradiation. Additionally, the NPs show excellent photothermal conversion. The researchers reported a significant synergistic effect of the photothermal therapy and chemotherapy, which was substantially higher than either therapy approach alone [98]. Another example is an even more complex nanoparticle combining five distinctly different functionalities [99]. PDA-coated, oleic acid-capped β-NaGdF_4_:Yb^3+^, Er^3+^@β-NaGdF_4_ NPs were used for T_1_-weighted magnetic resonance imaging, photothermal therapy, X-ray computed tomography, chemotherapy, and upconversion luminescence [99]. These NPs could completely eradicate a tumour in vivo, and the added functionalities allow for precise imaging possibilities [99].

PDA functionalisation on nanostructured lipid carriers (NLCs) was proposed to enhance NLC delivery in the skin. Chen et al. observed the formation of a PDA layer on NLCs using X-ray photoelectron spectroscopy and Fourier transform infrared spectroscopy (FTIR) [100]. Incorporating terbinafine as a model drug, an in vitro permeation study showed an increased delivery of terbinafine from NLCs to the deep skin layers, which was suggested to be controlled by the follicular pathway. In vitro cellular uptake using human immortalised keratinocytes (HaCaT) showed a higher uptake of PDA-coated NLCs without triggering additional cytotoxicity. By inhibiting endocytic routes, it was confirmed that the lipid raft/caveolae-mediated endocytosis was strongly involved in the internalisation of both the PDA-modified and unmodified NLCs.

Zeng et al. successfully synthesised hollow mesoporous MnO_2_ NPs loaded with photosensitiser chlorin e6 and further coated with folic acid-functionalised PDA as a functional platform for photothermal treatment and photodynamic therapy [101]. In an in vitro scenario, the nanocarriers not only allowed accurately controlled drug release and extensive oxygen production by reacting with endogenous H_2_O_2_ but also efficient NIR light-to-heat conversion, owing to the core–shell MnO2/PDA structure. Combining with the active target, enhanced permeability, and retention property of folic acid, a pronounced accumulation of nanocarriers at the tumour sites using an in vivo mouse model was observed. Upon irradiation with 660 nm and 808 nm, effective tumour growth inhibition was achieved, highlighting their prospective for improved cancer therapy and imaging.

PDA functionalisation on Cu^2+^-doped zeolitic imidazolate frameworks has been achieved by An et al. for glutathione-triggered and photothermal-reinforced sequential catalytic therapy against breast cancer [102]. In tumour microenvironments, the engineered NPs demonstrated a greater interaction with antioxidant glutathione (GSH), resulting in GSH depletion and Cu^+^ generation. The generated Cu^+^ would further catalyse local H_2_O_2_, allowing the production of highly toxic hydroxyl radicals (–OH) through an efficient Fenton-like reaction. It was further confirmed that the presence of PDA was able to yield high photothermal conversion effects, simultaneously accelerating GSH consumption and thus enhancing the Fenton-like reaction for further increases in intracellular oxidative stress.

Another example of PDA coating on porous structures such as mesoporous silica NPs loaded with a fluorescent dye was successfully achieved by Sapre et al. as an alternative drug delivery nanocarriers [103]. The particles were fed to Drosophila melanogaster to test their pH-responsive cargo release behaviour. The passage of the particles was monitored through the fly gut, and the result showed that the particles were decomposed inside the acidic environment of the middle midgut of the flies (pH < 4.0). However, in comparison to PEG-coated particles, PDA-coated particles exhibited lower specificity of release in the acidic middle midgut of flies and possessed a higher tendency to aggregate.

### 6.3. PDA Capsules

In addition to the coating, other formulations, such as PDA capsules, have been synthesised [86]. In general, the synthesis of polymer capsules is rather complex, work-intensive, and therefore expensive [104]. PDA capsules, on the other hand, are comparatively easy to produce [105]. Moreover, the unique properties of PDA combined with drug loading and the simultaneous possibility of ligand grafting add to the benefit of creating capsules of the material. PDA capsules are mainly formed using templates [106]. In general, a template nanostructure is coated with PDA, and subsequently, the template is removed by optimised methods. Various modifications of this approach have been reported to date. For instance, CaCO_3_ microparticles, silica NPs, MnCO_3_ particles, and polystyrene spheres have recently been used as templates to produce PDA capsules of different shapes and sizes [105].

#### 6.3.1. Biocompatibility of PDA Capsules

These PDA capsules were, as expected, found to not have any significant impact on cell viability in vitro, very similar to the previously described coated NPs [107]. Even though PDA capsules have not yet been tested in an in vivo study, the data that is available for PDA-coated NPs and also PDA NPs (see following chapter) suggest that they are safe for relevant biomedical applications.

#### 6.3.2. Biomedical Applications of PDA Capsules

Postma et al. produced PDA-based capsules and loaded or bonded them with functional drugs for drug delivery [107]. Doxorubicin-containing capsules were reported to kill HeLa cancer cells in vitro more efficiently than free doxorubicin at the same concentration [107]. In another approach, Deng and co-workers used various sacrificial templates (e.g., SiO_2_ colloidal NPs and nanorods and sulfonated polystyrene microspheres) to produce hollow PDA NPs with the same shape [108]. This was achieved by coating the templates with a PDA layer and subsequently dissolving the underlying nanoparticle thermally [108]. Moreover, the resulting PDA structure surface was modified with different moieties or heat-resistant cargo (previously immersed into the template NPs) and loaded into the hollow PDA NPs [108]. Meng and co-workers also produced PDA capsules using a sacrificial template of SiO_2_ colloidal NPs [109]. The researchers loaded the capsules with ionic liquids and used them for microwave thermal therapy to destroy tumour tissue [109]. Upon intravenous injection with the subsequent treatment of a single dose of microwave irradiation, encouraging antitumour effects in mice were observed (Figure 6) [109]. While there are a variety of approaches to synthesis and up-scale PDA capsules, their benefit and interactions in complex physiological media in biological systems need further investigations.

PDA encapsulation of fullerenes (C60) and GSH via the Michael-addition reaction possessing reactive oxygen species (ROS) scavenging properties was achieved by Zhang et al. [110]. The electron affinity of fullerenes facilitates the carbon structures to eliminate ROS, which often leads to toxicity or biological dysfunction. In vitro experiments using human epidermal keratinocytes (HEK-a), human umbilical vein endothelial cells (HUVEC), human microglia (HM), and normal liver cells (L-02) cells observed enhanced biocompatibility and cytoprotective roles against oxidative stress induced by H_2_O_2_ in these human cells at low concentrations (2 μg/mL).

Another example of PDA encapsulation on NPs was demonstrated by Liu et al. [111]. In this case, polymeric NPs (fabricated from the star-shaped copolymer cholic acid-poly(D,L-lactide-co-glycolide) (CA-PLGA)) and loaded with hydrophobic anticancer drug docetaxel (DTX) were coated with PDA, followed by the conjugation of poly(elethyl glycol) (PEG)-modified targeting ligand aptamer AS1411 (Apt) and adsorption of the hydrophilic anticancer drug doxorubicin. This “four-in-one” nanoplatform exhibited high NIR photothermal conversion efficiency and pH and thermoresponsive drug release behaviour. Moreover, it was able to specifically target human breast carcinoma cells (MCF-7), whilst delivering synergistic chemo-photothermal treatment, leading to improvement of the anticancer effect both in vitro and in vivo.

### 6.4. PDA NPs and Nanocomposites

As previously mentioned, PDA has the ability to efficiently convert light into heat [20]. By designing PDA NPs, an efficient new platform for photothermal therapy is envisaged. Due to additional favourable features (e.g., biocompatibility and the possibility for π–π stacking), the NPs can also be used for biomedical applications such as drug delivery or imaging [75]. Depending on the required application and/or size, different approaches to produce PDA NPs are described [106]. One of the more widely used methods is based on the oxidative self-polymerisation of dopamine in an ethanol/water mix with slight modifications to the reaction (i.e., pH and reaction time) [9,112,113]. NPs in a size range between 70 and 400 nm have been produced by this reaction, whereas the ratios of ethanol to water as well as the concentration of dopamine influences the size of the produced NPs [114]. Another approach is the mixture of dopamine and protein solutions in TRIS-buffered water at pH 8.5 [23]. The size can be tuned by the ratio of dopamine and protein concentrations, having the added benefit of directly incorporating functional proteins into the PDA NPs [23].

#### 6.4.1. Biocompatibility of PDA NPs

Similar to their resultant coatings and capsules, PDA NPs were also found to have no adverse effects on cell viability upon cell internalisation [9]. Zhang et al. exposed mouse fibroblasts to increasing concentrations of PDA particles (10–160 µg/mL) and did not report any adverse effects [115]. Hybrid PDA NPs containing only human serum albumin (HSA) or bovine serum albumin (BSA) and transferrin were synthesised and exposed to human fibroblasts, mouse melanoma, and mouse macrophages [23,35]. None of the conditions or concentrations (5–160 µg/mL) showed any negative effects on cell growth [23,35]. Even though considerable amounts of data investigating the biocompatibility of PDA exist, not very much is known about the fate of the material. In acidic milieu, PDA has been reported to degrade, and thus the material is believed to dissolve inside the acidic environment present in lysosomes [116,117,118]. In biological environments, this has mainly been demonstrated on grounds of the shedding and/or dissolution of PDA coatings in vitro [116,117]. For instance, Ding et al. have used mesoporous silica NPs coated in PDA and reported the peeling of the PDA layer under acidic conditions [118]. In vivo studies with PDA NPs reported the intravenous LD_50_ for rats to be at 483.95 mg/kg [9]. This LD_50_ is around 4.5 times higher than for caffeine (105 mg/kg) and 150 times higher than that for 1.9 nm gold NPs (3.2 mg/kg) [119,120]. After a single injection of PDA NPs, neither a significant change in biochemical indicators for liver function nor any appreciable signs of morphological change, fibrosis, or inflammation in a selection of organs (i.e., kidney, spleen, lung, liver, and heart) were found [9]. Based on the available data, PDA can be considered as biocompatible to the greatest possible extent and can readily be investigated further for applications in medicine and other fields.

#### 6.4.2. Biomedical Applications of PDA NPs

PDA NPs were first employed by Liu et al. as a phototherapeutic agent in mice (Figure 7A) [9]. The researchers injected PDA NPs into a tumour graft and subsequently irradiated the tumour area with laser light, resulting in a complete tumour ablation after two days (Figure 7B,C) [9]. Our group modified the NPs to add targeting properties to improve the uptake into the endolysosomal system in cancer cells [35]. To this end, PDA NPs containing transferrin as the targeting moiety were synthesised [35]. By irradiating mouse melanoma cells in vitro that were previously exposed to the NPs, cell death could be induced mediated by lysosomal membrane permeabilisation [35]. PDA NPs also allow for a direct loading of hydrophobic molecules [121]. The incorporation of aromatic molecules is supposed to be driven by the π–π stacking present in the material [122]. It is hypothesised that these aromatic compounds can intercalate into the PDA structure [122]. Ho and Ding prepared PDA NPs loaded with camptothecin which, when put in phosphate-buffered saline (PBS), gradually released the drug [112]. The camptothecin-loaded PDA NPs were found to inhibit cell growth to a similar degree as free camptothecin [112]. Li et al. loaded doxorubicin directly into PDA NPs and used them for combined photothermal and chemotherapeutic therapy [123]. The NPs were reported to deliver their cargo into the cancer cells and, upon near-infrared irradiation, released their payload and efficiently killed the cells [123]. Similarly, Poinard et al. incorporated the hydrophobic photosensitiser drug chlorin e6 into PDA NPs [124]. When irradiated with 665 nm light, the drug was released and, thus, photodynamic and thermal therapy were successfully combined with drug delivery [124]. Zhou et al. were also able to incorporate a dye into PDA NPs and subsequently functionalise them with a targeting moiety [125]. Surface-enhanced Raman scattering and fluorescence microscopy revealed the NPs to be located inside the cells [125]. Dong and co-workers combined doxorubicin-loaded PDA NPs with a targeting function and photothermal therapy, also resulting in a synergistic effect [126].

In an attempt to combine chemo and photothermal therapies, Cheng and co-workers designed PEG-modified PDA NPs [113]. The PDA–PEG NPs were loaded with doxorubicin and the synergistic effects of the two therapeutic approaches were reported in vitro and in vivo [113]. There is also a number of publications reporting the applicability of PDA NPs for theranostic and bioimaging applications. In terms of bioimaging, the broad absorption spectrum and connected excellent photoacoustic signal allows PDA NPs to be used directly as a bioimaging agents in vivo [127]. Ju et al. even aimed to enhance the signal in the near-infrared part of the absorption spectrum by introducing hydrolysis-sensitive citraconic amide to the PDA nanoparticle [128]. The hydrolysis distorts the positive surface charge of the nanoparticle, resulting in aggregation and therefore an enhanced photoacoustic signal in comparison to unmodified PDA NPs [128]. The hydrolysis of these modified NPs was reported to increase the photoacoustic signal up to eight-fold [128], and combined photothermal therapy with photoacoustic imaging and drug delivery was possible [123].

Additionally, a targeting moiety (i.e., arginine–glycine–aspartic acid–cysteine peptide) combined with drug loading such as doxorubicin on the PDA particle was successfully introduced [123]. The synthesised NPs were able to deliver the loaded doxorubicin to the targeted cancer cells both in vivo and in vitro by a combination of pH and near-infra red stimuli [123]. In the bioimaging field, Liu et al. presented a different method to synthesise PDA NPs for magnetic resonance imaging [129]. The method allows for a synthesis of smaller PDA NPs, which the authors subsequently conjugated with ferric ions (Fe^3+^) [129]. Upon pH change, these NPs displayed magnetic resonance imaging contrast as well as high photothermal performance. Subsequent tumour growth in vivo was completely inhibited using photothermal therapy guided by magnetic resonance imaging [129]. Cai and co-workers also developed a PDA-based nanoparticle for magnetic resonance and photoacoustic imaging as well as photothermal therapy [130]. They achieved this by adding iron ions to the PDA NPs, which are known to be coordinated by PDA and also indocyanine green to enhance PDA’s absorption in the near-infrared part of the spectrum [8,130]. These NPs are highly efficient for photothermal treatments while also being usable for photoacoustic and magnetic resonance imaging in vivo [130]. In an effort to combine radioisotope therapy and chemotherapy, Zhong et al. loaded PDA NPs with ^99m^Tc and ^131^I as well as doxorubicin [131]. In vivo studies showed an excellent synergistic effect whilst not increasing the toxicity for the treated animals was seen [131].

Recently, photothermal treatment application based on hybrid PDA NPs consisting of l-arginine (l-Arg), indocyanine green (ICG), and mesoporous PDA (MPDA) was designed by Yuan et al. as an alternative to eliminate bacterial biofilm (Figure 8A) [132]. The synthesised particles displayed relatively uniform size distribution with a diameter of about 200 nm (Figure 8B), optimal ICG loading efficiency of more than 90.0% was achieved, and the loading content was 9.0 μg (Figure 8C–E). It was demonstrated that upon exposure to near-infrared (NIR) light, the engineered nanoplatform was not only able to generate heat but also produced ROS, causing a cascade catalysis of l-Arg to release nitric oxide (NO). The authors suggested that biofilm destruction was associated to the NO-enhanced photodynamic therapy and low-temperature photothermal treatment (≤45 °C), causing severe destruction of the membranes of bacteria.

Feng et al. successfully synthesised a multifunctional zeolitic imidazolate frameworks 8 (ZIF-8)-gated PDA NP carrier for simultaneously delivering a photosensitiser and a catalase (CAT) into tumour cells [133]. The ZIF-8 gatekeeper enables the concurrent and effective delivery of these functional payloads and the successive tumour acidic pH-stimulated drug release. This leads to a substantial improvement of combination efficacy by increasing tumour hypoxic conditions, since the CAT-mediated O_2_ generation could substantially promote an efficient photodynamic therapy operation. Moreover, the ability of the nanoplatform to effectively convert NIR light into heat could result in the thermal elimination of tumours.

A dopamine–melatonin nanocomposite (DM-NC) possessing a synergistic NIR responsive photothermal treatment and pharmacological modality was designed by Srivastava et al. to target a short Amyloid-β (Aβ) peptide that is responsible for Alzheimer’s disease (AD) [134]. The non-covalent interaction-mediated self-assembly of melatonin and dopamine oxidative intermediates leads to the evolution of DM-NCs that can interestingly resist the changing of pH and peroxide environment. The NIR-activated melatonin production and photothermal effect collectively prevent Aβ nucleation, self-seeding, and propagation as well as disrupt the preformed Aβ fibers. This nanocomposite exhibited a lower toxicity to neuroblastoma cells in vitro, could suppress the AD-associated generation of intracellular reactive oxygen species, and is devoid of any negative impact on the axonal growth process. Using okadaic acid-induced neuroblastoma and ex vivo midbrain slice culture-based AD model, nanocomposite exposure shown to suppress the intracellular Aβ production, aggregation, and accumulation, demonstrating a potential application of multimodal NIR-responsive synergistic photothermal treatment and pharmacological modality of DM-NCs for effective AD therapy.

## 7. Conclusions and Outlook

The use of PDA in various applications including biomedical is an excellent example of bioinspiration and the successful translation of effects observed in nature. The adhesive properties of dopamine as observed in mussels has inspired the use of the material as a molecular glue, thereby adding new or enhancing the existing properties of the coated material. Approaches to design heat-inducible PDA NPs for theranostic applications also emerged due to its high biocompatibility and relatively easy system to produce. Additionally, PDA provides a promising platform for imaging, i.e., magnetic resonance, photoacoustic, computer tomography, and fluorescence, whether as a nanoparticle, nanocomposite, or as a coating of existing materials. Nonetheless, a thorough investigation on the mechanisms of PDA formation should be performed systematically in order to provide enough knowledge that allows us to be able to fully control the physicochemical features and structure–function relationships. A large-scale synthesis of PDA will be another challenge to tackle, as most of the aforementioned examples shown were performed in small laboratory environments. Further development of the PDA nanocomposite should also be emphasised in a (tumour) targeting aspect without sacrificing the therapy and diagnostic features of PDA. The immune response of PDA particles once injected to human/animal bodies should be also examined. More applications in brain-related fields should be designed as dopamine (a neurotransmitter and precursor of PDA) is present in our nervous system.

The field of PDA-based nanomaterials is in its early stages, but the existing examples and applications have shown their potential. Their biocompatibility, degradation in acidic environments combined with drug delivery, imaging, and photothermal properties can provide promise for its approval for future clinical application.

## Figures and Tables

**Figure 1 materials-13-01730-f001:**
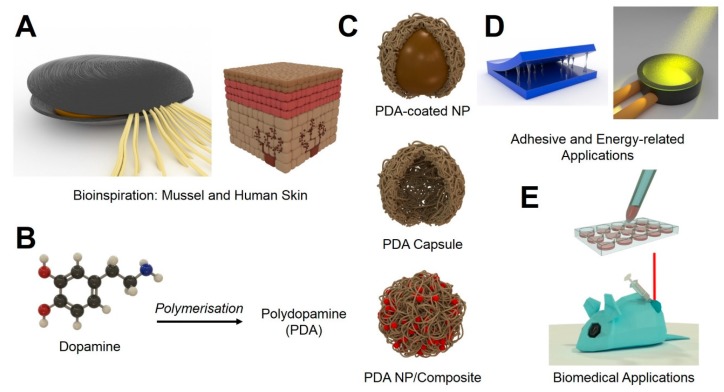
Bioinspiration from mussel and human skin (**A**). Formation of polydopamine (PDA) from polymerisation of dopamine molecule (**B**). Illustration of formation different types of PDA structures, namely, PDA-coated NP, PDA capsule, and PDA NP/composite (**C**), their potential use as adhesive and in energy-related applications (**D**) and biomedical application (i.e., drug delivery and photothermal therapy) (**E**).

**Figure 2 materials-13-01730-f002:**
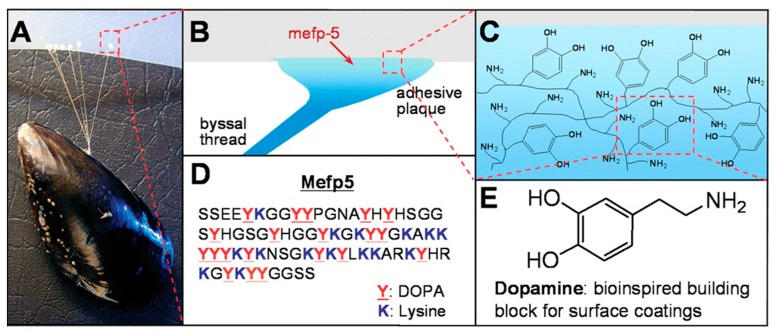
Mussel (*mytilus edulis*) attached to poly(tetrafluoroethylene) (**A**). Simplified schematic illustration of the interface between mussel byssi and substrate (**B**). Simplified molecular structure of PDA depicting the amine and catechol groups (**C**). Amino acid sequence of mytilus edulis foot protein 5 (**D**). Lewis formula of dopamine (**E**). From reference [17]. Reprinted with permission from the American Association for the Advancement of Science (AAAS).

**Figure 3 materials-13-01730-f003:**
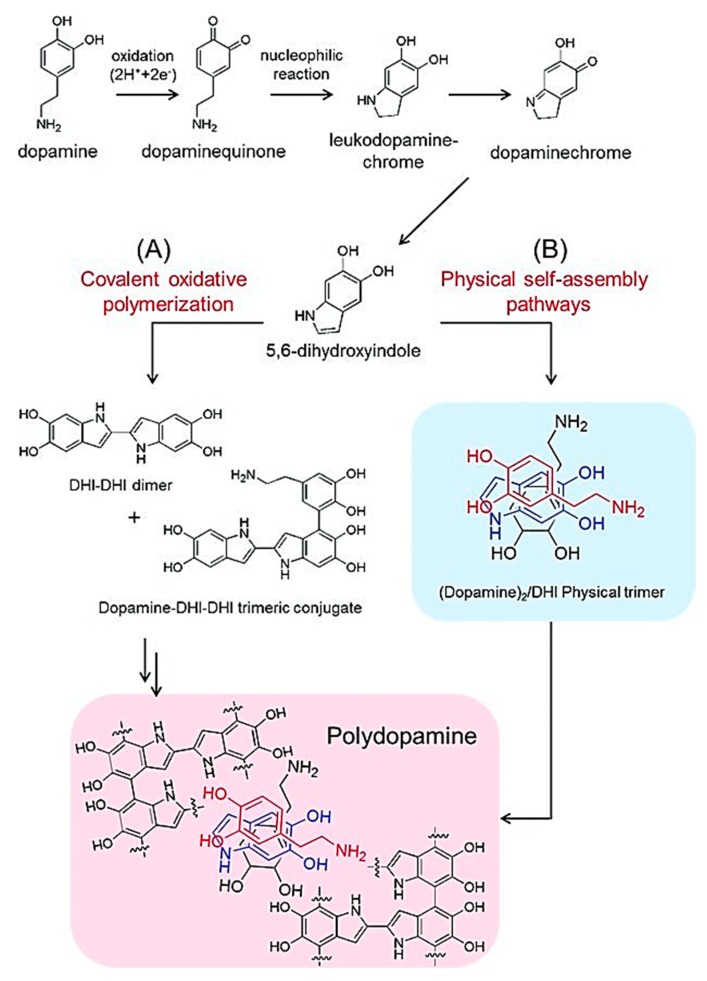
Schematic representation of two pathways hypothesised to occur simultaneously in PDA synthesis. Oxidative polymerisation forming covalent bonds (**A**) and physical self-assembly of reaction intermediates and dopamine (**B**). From reference [42]. Reprinted with permission from Wiley.

**Figure 4 materials-13-01730-f004:**
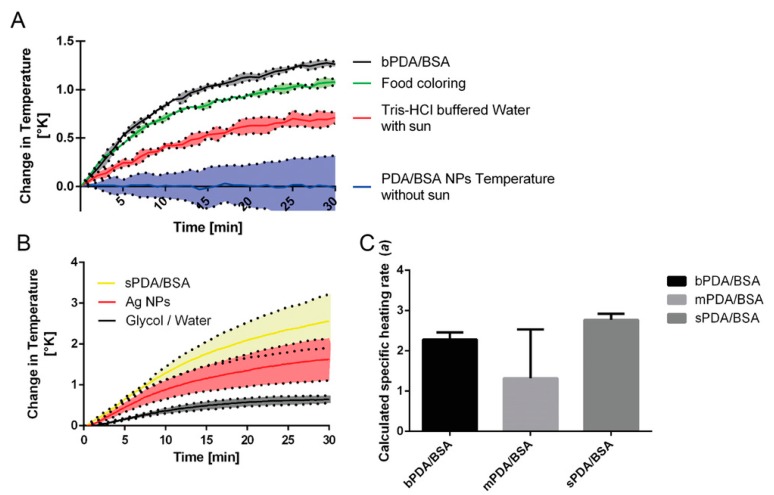
‘Big’ PDA NPs containing BSA (bPDA/BSA, black line) exhibited superior heating abilities of the solar fluid in comparison to micron-sized soot particles (food coloring, green line) and solar fluid (TRIS-HCl-buffered water) without any addition (red line) (**A**). ‘Small’ PDA/BSA NPs (sPDA/BSA, yellow line) exhibited an increased heating ability of the solar fluid in comparison to silver NPs (Ag NPs, red line) and plain solar fluid (glycol/water, black line) (**B**). Calculated specific heating rate of three different sizes of PDA NPs containing BSA: ‘big’ PDA/BSA NPs (bPDA/BSA), ‘medium’ PDA/BSA NPs (mPDA/BSA), and ‘small’ PDA/BSA NPs (sPDA/BSA) (**C**). From reference [59]. Reprinted with permission from Wiley.

**Figure 5 materials-13-01730-f005:**
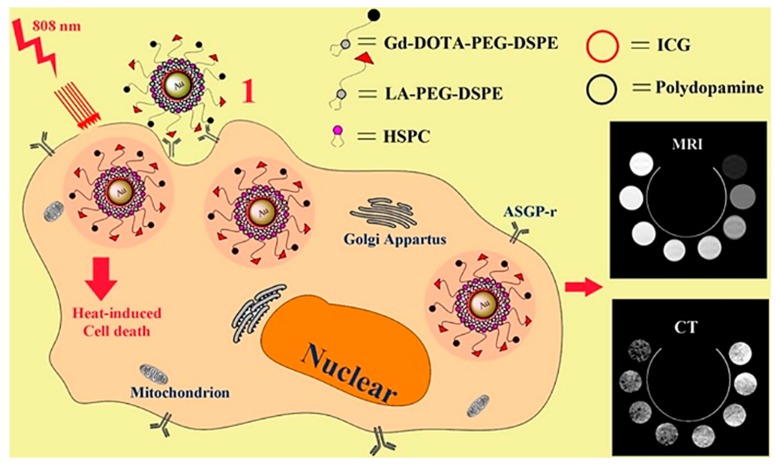
Gold NPs which were coated with PDA, indocyanine green, and modified lipids (gadolinium-1,4,7,10-tetraacetic acid and lactobionic acid) were used for targeted magnetic resonance imaging, computed X-ray tomography, and photothermal therapy. Reprinted with permission from [93]. Copyright 2020 American Chemical Society.

**Figure 6 materials-13-01730-f006:**
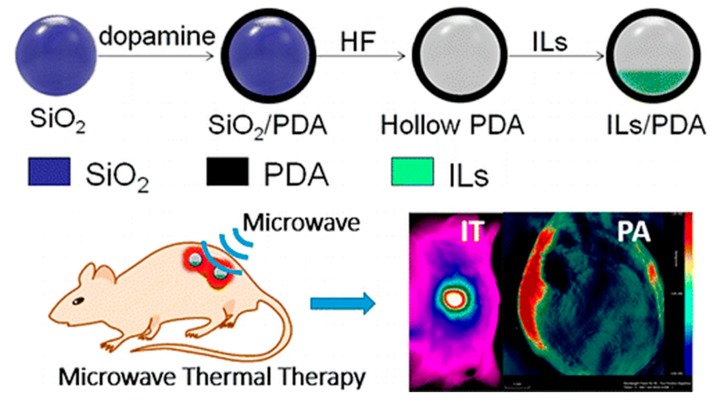
PDA capsules produced using a sacrificial SiO_2_ template were filled with ionic liquids. In vivo, the capsules showed excellent efficiency for microwave heating. Reprinted with permission from [109]. Copyright 2020 American Chemical Society.

**Figure 7 materials-13-01730-f007:**
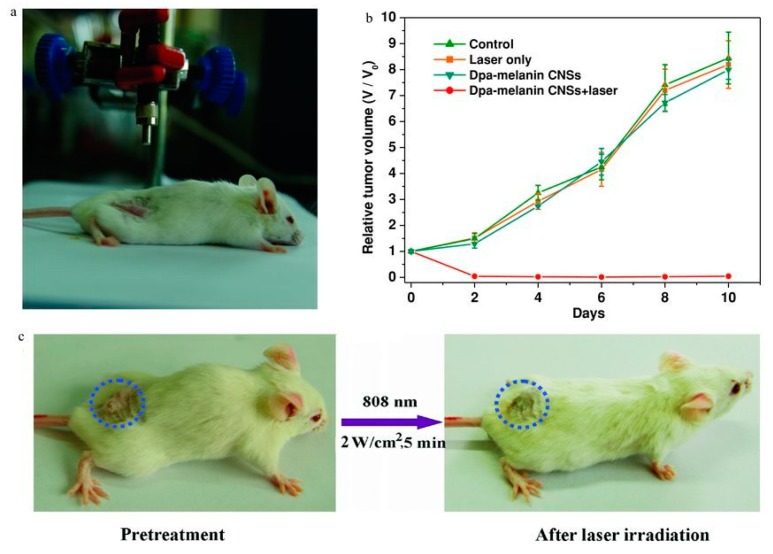
Laser set up for photothermal therapy of the tumour-bearing mouse (**a**). Relative tumour volume over 10 days comparing the different treatments: Control (green line), laser only (orange line), PDA NPs alone (blue line), and PDA NPs irradiated with laser light (red line) (**b**). Photos of a tumour-bearing mouse before and after treatment with the laser set up (**c**). From reference [9]. Reprinted with permission from Wiley.

**Figure 8 materials-13-01730-f008:**
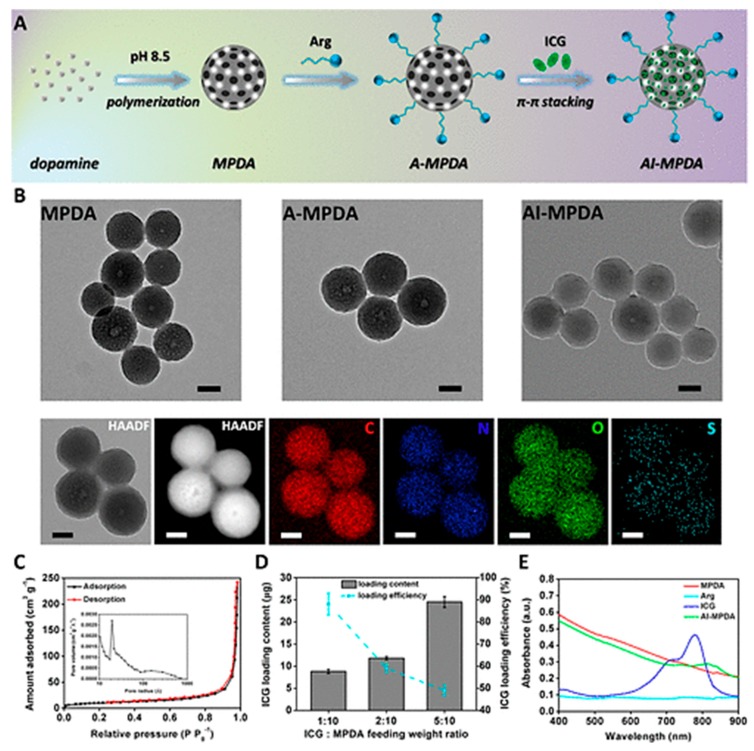
(**A**) Schematic illustration showing the preparation of hybrid PDA NPs consisting of l-arginine (l-Arg, A), indocyanine green (ICG, I), and mesoporous PDA (MPDA). (**B**) Representative TEM micrographs of different NPs, high-angle annular dark-field scanning transmission electron microscopy image of hybrid PDA NPs, and corresponding elemental mapping (scale bar: 100 nm). (**C**) N_2_ adsorption/desorption isotherms and pore-size distribution (inset) of MPDA NPs. (**D**) ICG loading content and efficiency in AI-MPDA NPs at different ICG:A-MPDA feeding weight ratios (0.1 mg of A-MPDA). (**E**) UV/vis absorbance spectra of MPDA (0.1 mg mL^−1^), Arg, ICG (5 μg mL^−1^), and AI-MPDA NPs (0.1 mg mL^−1^). From reference [132] Reprinted with permission from the American Chemical Society.

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
