# Peer review of "From Bioinspired Glue to Medicine: Polydopamine as a Biomedical Material"

_materials, 2020, doi:10.3390/ma13071730_

Round 1

Reviewer 1 Report

Hauser et al review the the development of PDA in the field of biomedical application. This review is well organized from the PDA struture, optical properties to their applications. I recommend it publication in the present form.

Reviewer 2 Report

Rothen-Rutishauser et al. review some of the recent investigations and applications of polydopamine (PDA) as a biomedical material. There has been quite a significant research effort in studying PDA for a variety of applications. Some of these investigations have led to very important discoveries and to advancing the field. PDA is widely used, and its use for biomedical and other applications has been reviewed by others. For example, a recent review by Zhao et al (ACS Nano 2019, 13, 8, 8537-8565).

I have concerns about what is this review offering that is not available in current the literature. I have noticed a very limited number of works published in the last year reviewed here. This could add some novelty to the work, which at the moment is lacking. Also, I see repeated references to Review papers, for example reference [8]. Referencing a review and not to the original work may be misleading, and does not add anything to that already published work.

The authors give a brief introduction on current research trend around PDA, and relate this to the advances in nanotechnology. It is a very brief introduction, where the reader cannot grasp the idea of what is going to be reviewed thereafter. I suggest this should be further developed. The ‘Bioinspiration by mussels’ and ‘Bioinspiration by human skin’ is appropriate.

Another concerning section is that related to the ‘Adhesive and energy-related applications’. It is difficult to understand how this is related to biomedical applications, or relevant to the current review. It is confusing and distracting, in my point of view.

Then the authors (in section 5) review the current biomedical applications of PDA as an adhesive, coating of nanomaterials, capsules and nanoparticles. Each of these sections dedicate a paragraph to discuss the biocompatibility of the material/structure. This seems repetitive at times, and I wonder if the biocompatiblity of PDA in its different forms could be addressed. As mentioned before, I am missing references to newer research and a deeper discussion on

Finally, the conclusions and outlook section should be improved. The authors should provide a greater insight in prospective research, and how researchers may overcome the current challenges that the field is facing to translate laboratory research to real-life medical applications.

Reviewer 3 Report

The manuscript "From Bioinspired Glue to Medicine: Polydopamine as a Biomedical Material " by Rothen-Rutishauser and coworkers summarizes the literature regarding polydopamine in the biomedical field. The review is well-written but a minor spell check is required. It should be published after minor revisions.

General comments:

  1. I suggest to add an additional scheme to show the content of the review.
  2. I suggest to add 1-2 additional Figures.
  3. How does PDA coating effect the blood circulation of delivery vehicles?
  4. I think the authors should try to improve the connection between the Figures and the text.

5.The conclusions and outlook Section is quite short. It should be expanded. What are the challenges? What are possible directions? What is missing in the literature?

  1. The formatting of Chapter 5 with sections and subsections looks a bit strange. Maybe introduce 5.1 etc?
  2. The authors should check the following literature, which might give some additional input:

https://pubs.rsc.org/en/content/articlelanding/2020/py/d0py00085j

https://pubs.acs.org/doi/abs/10.1021/acs.biomac.8b01301

https://link.springer.com/article/10.1007/s11426-018-9392-6

Specific comments:

  1. SPION is not explained

Reviewer 4 Report

The work is very interesting and easy understanding for readers. I recommend an extensive English language revision before publication.
A lot of typos errors should be corrected.

Round 2

Reviewer 2 Report

The authors have addressed most of the concerns raised and the review has significantly improved. 

However, before this reviewer recommends it for publication, my concern around repeatedly referencing another reviews needs to be addressed. 

Specifically, I noticed ~16 references to work [8] Liu et al, Chem Rev, 2014, 114, 5057. This almost looks like rephrasing what others have already excellently reviewed! The authors need to reference the original works - and not the review - if this is relevant to their paper. Otherwise the value of the present work is greatly reduced.

Author Response

We appreciate the comment. Hence, where possible, the original works have been referenced. The changed references have been marked in the re-submitted manuscript in green.